# Integrins Can Act as Suppressors of Ras-Mediated Oncogenesis in the *Drosophila* Wing Disc Epithelium

**DOI:** 10.3390/cancers15225432

**Published:** 2023-11-15

**Authors:** Ana Martínez-Abarca Millán, María D. Martín-Bermudo

**Affiliations:** Centro Andaluz de Biología del Desarrollo, CSIC, Universidad Pablo de Olavide, 41013 Sevilla, Spain

**Keywords:** *Drosophila* integrins, cancer, cell growth, invasion

## Abstract

**Simple Summary:**

Studies over the last few years have revealed that integrin function in cancer is controversial. Accumulating evidence has shown that integrins can act as both tumour promoters or tumour suppressors. The targeting of integrins has shown enormous potential in both the diagnosis and treatment of cancer. Thus, further mechanistic insights into the roles of integrins as regulators of cancer progression are required. In this work, we aim to use the *Drosophila* model to gain insight into the role of these extracellular matrix receptors in tumoural progression. We find that the role of integrins as tumour suppressors is conserved. The depletion of integrins enhances the growth and invasive behaviour of tumours induced by a gain of function of an oncogenic form of Ras, by regulating cell shape and cycle progression. Furthermore, our results show that integrin loss enhances the ability of Ras tumour cells to affect the tumour microenvironment through the activation of the JNK pathway.

**Abstract:**

Cancer is the second leading cause of death worldwide. Key to cancer initiation and progression is the crosstalk between cancer cells and their microenvironment. The extracellular matrix (ECM) is a major component of the tumour microenvironment and integrins, main cell-ECM adhesion receptors, are involved in every step of cancer progression. However, accumulating evidence has shown that integrins can act as tumour promoters but also as tumour suppressor factors, revealing that the biological roles of integrins in cancer are complex. This incites a better understating of integrin function in cancer progression. To achieve this goal, simple model organisms, such as *Drosophila*, offer great potential to unravel underlying conceptual principles. Here, we find that in the *Drosophila* wing disc epithelium the βPS integrins act as suppressors of tumours induced by a gain of function of the oncogenic form of *Ras*, *Ras^V^*^12^. We show that βPS integrin depletion enhances the growth, delamination and invasive behaviour of *Ras^V^*^12^ tumour cells, as well as their ability to affect the tumour microenvironment. These results strongly suggest that integrin function as tumour suppressors might be evolutionarily conserved. *Drosophila* can be used to understand the complex tumour modulating activities conferred by integrins, thus facilitating drug development.

## 1. Introduction

Cancer is the second leading cause of death worldwide. In 2020, there were 19.3 million new cancer cases, and by 2040, this number is expected to rise to 29.5 million [1]. Thus, cancer research studies are crucial in diminishing the tremendous human and economic tolls of cancer.

The crosstalk between cancer cells and their microenvironment plays a key role in cancer initiation and progression. The extracellular matrix (ECM) is a major component of the tumour microenvironment and integrins, main cell adhesion receptors for components of the ECM, are involved in almost every step of cancer progression, including cancer initiation, proliferation, metastasis and survival of circulating tumour cells (reviewed in [2]). Integrins are heterodimeric molecules containing an α and a β subunit. The mammalian family consists of twenty-four heterodimers generated from a combination of eighteen α integrin and eight β integrin subunits (reviewed in [3]). Much attention has been devoted to the β1 family of integrins. These studies have demonstrated that the biological roles of β1 integrins in cancer, as signalling molecules, mechanotransducers and essential components of the cell migration machinery, are quite complex and highly dependent on the type and developmental stage of the tumour. In different types of cancers, the expression of β1 integrin facilitates the growth and survival of cancer cells. However, regarding metastasis, the effects of β1 integrin are controversial and β1 integrins can act as tumour promoters or tumour suppressors. For example, while most β1 integrins, and more specifically α3β1, are essential for mammary tumorigenesis [4,5], α2β1 integrin acts as a metastasis suppressor in breast cancer [6]. Furthermore, when cultured in three-dimensional ECM scaffolds, the block or knockdown of the β1 integrin function elicits a pro-metastatic switch in human and mouse E-cadherin-positive triple-negative breast cancer cells [7]. Thus, given the complexity of β1 integrins and their antagonistic roles in some types of cancer cells, therapeutic targeting of these receptors is a challenge and only a small number of clinical trials have been successful. This incites a better understating of the role of integrins in cancer progression, which could lead to the development of new targeting approaches. To achieve this goal, simple model organisms, such as *Drosophila*, with fewer genes compared to the human genome and lower genetic redundancy, have great potential to unravel underlying conceptual principles.

In contrast to vertebrates, the *Drosophila* genome encodes only two β subunits, βPS and βν, and five α subunits, αPS1 to αPS5. The βPS subunit, encoded by the gene *myospheroid* (*mys*), the orthologue of vertebrate β1, is widely expressed and forms heterodimers with all α-subunits (reviewed in [8,9]). The βPS integrins have been shown to regulate many cellular processes both during embryonic and larval development and in the adult. In particular, the primordium of the *Drosophila* wing, the larval wing imaginal disc has been successfully used to decipher the function of integrins during epithelial morphogenesis and homeostasis (reviewed in [10,11,12,13,14]). The formation of the wing begins during early embryonic development, when about 30 cells are assigned to build the primordium of the wing imaginal disc (henceforth the wing disc). During larval life, disc cells proliferate and form an epithelial sac, comprised of a squamous layer of epithelial cells, named the peripodial epithelium (PE), and a columnar epithelial layer termed the disc proper, to which myoblasts, tracheal cells and a few neurons attach. Underlying the epithelial cells of the disc proper there is a basement membrane (BM). The wing disc epithelium is divided into four broad domains, the pouch, the hinge, the notum and the PE. The pouch and the hinge will produce the wing blade and wing hinge, respectively, while the notum will give rise to most of the back of the fly in the thorax, and the PE, which gives rise to some of the pleura (reviewed in [15]). Integrins have been shown to be required for the adhesion, shape and survival of the wing disc epithelial cells [14,16,17,18]. The wing imaginal disc also constitutes an excellent and productive choice for investigating the biology of carcinoma, the most common cancer type that originates from epithelial cells (reviewed in [19]). More specifically, the wing imaginal disc has been extensively employed to study the progression of tumours due to mutations in the proto-oncogene *Ras* [20], mutated in 30% of human cancers [21]. The induction of groups of cells over-expressing an oncogenic-activated form of *Ras* (*Ras^V^*^12^) in the wing disc results in hyperplastic growth. This has been used in genetic screens to isolate additional genes, which can act as either suppressors or enhancers of the growth of *Ras^V^*^12^ overexpressing cells [20]. In this context, we have recently found that the elimination of integrins from transformed *Ras^V^*^12^ wing disc epithelial cells stimulates their basal extrusion [14]. However, little more is known about the role of the βPS integrins in cancer initiation and progression.

In this work, we use the wing disc to analyse the role of integrins on the progression of *Ras^V^*^12^ epithelial tumour cells. We find that downregulation of the βPS integrins enhances *Ras^V^*^12^-mediated tissue hyperplasia. The elimination of integrins promotes the growth of *Ras^V^*^12^ tumour cells. In addition, using live imaging and confocal microscopy, we find that the reduction of βPS integrin levels induces the delamination and invasion of tumour cells to neighbouring wild-type regions. These results demonstrate that, as is the case in some mouse models and human cancer cell lines, in *Drosophila*, integrins inhibit the metastatic capacity of tumour cells. These results also strongly suggest that the role of the β1 integrin family as suppressors of tumour metastasis might be evolutionarily conserved.

## 2. Materials and Methods

### 2.1. Fly Strains

The following stocks were used: *UAS-Ras^V^*^12^ [22]; *Ap>*myrT/Cyo, *Ap>*GFP/Cyo, *Ap-Gal4-UAS-flyfucci*/Cyo (gifts from Dr. Marco Milán’s laboratory); *UAS-mys^RNAi^* and *UAS*-*hid^RNAi^* (Vienna Drosophila Resource Center, Vienna, Austria). Flies were raised at 25 °C.

### 2.2. Immunocytochemistry and Imaging

Third instar larvae wing imaginal discs were dissected and stained following standard procedures and mounted in Vectashield (Vector Laboratories, Burlingame, CA, USA). The primary antibodies used were: goat anti-GFPFICT (Abcam, Cambridge, UK, 1:500), rabbit anti-caspase Dcp1 (Cell Signaling, Danvers, MA, USA; 1:100), rabbit anti-pJNK (Promega, Madison, WI, USA, 1:200), mouse anti-βPS (Developmental Studies Hybridoma Bank, DSHB, University of Iowa, United States, 1:50), rabbit anti-PH3 (EMD Millipore, Burlington, MA, USA, 1:250), rabbit anti-Perlecan (Dr. A. González-Reyes, 1:850), rabbit anti-aPKC (Promega, 1:200), Rat-anti-RFP (Chromotek, Planneg, Germany, 1:500). The secondary antibodies employed were Alexa fluor 488, (Molecular ProbesTM, Eugene, OR, USA) and Cy3 and Cy5 (Jackson ImmunoReseach Laboratories, Inc., West Grove, PA, USA) at 1: 200. DNA was labelled using Hoechst (Molecular Probes, 1:1000). F-actin was visualized using Rhodamine Phalloidin (Molecular Probes, 1:50).

Confocal images were obtained using a Leica SPE microscope, equipped with a 20× oil objective (NA 0.7) and a Leica Stellaris microscope equipped with 20×, 40× and 63× oil objectives with NA 0.75, 1.4 and 1.4, respectively.

For live imaging, wing disc samples were prepared as described in [23] and filmed for 2–3 h using a Leica Stellaris microscope and a 40× oil objective.

### 2.3. Quantification of Fluorescence Intensity

To measure the fluorescent intensity of all markers, fluorescent signalling was quantified on several confocal images per genotype, using the square tool FIJI-Image J version 2.0.0-rc-59/1.51n. The microscope settings, such as the laser parameters and z-confocal interval, were maintained between imaging sessions in each experiment. Measurements of whole fluorescence intensity were carried out using the “mean grey value” tool in FIJI. This tool divides the mean of all included pixel intensities by the outlined cell area. The different areas of the wing disc, apterous (dorsal) and control (ventral) domains were selected using the wand tool.

To quantify cell proliferation, dots of the fluorescent intensity of wing discs stained with an anti-PH3 antibody were quantified using the FIJI-Image J tool to analyse particles. Images were previously processed using the tool threshold, which transforms 8-bit images into a binary system.

Cell areas were calculated by manually delineating the apical and basal surfaces of the cells and measuring the resulting surfaces using FIJI-ImageJ software. To quantify cell height, a vertical line from the apical to the basal surface was drawn in a region of interest of a wing disc stained for F-actin, and the length of this line was measured using FIJI.

Flyfucci analyses were performed using the “merge channels” tool in FIJI to generate an image containing the green and red channels together. The “TrackMate” tool was used to measure the fluorescence intensity of both channels at the same 3D coordinates. Spots showing similar levels of the fluorescence intensity of both channels correspond to G2 and those with higher levels, at least three times greater, in the red channel with respect to the green one and vice versa correspond to S and G1, respectively.

All graphics and mathematical analysis were generated using R software and the ggplot2 package. For statistical comparisons, the *p*-value was calculated using the Welch test and assuming that all distributions were normal with unequal variances. Multiple comparison *p*-values were adjusted following the BH procedure (controlled False Discovery Rate).

## 3. Results

### 3.1. Integrin Knockdown Enhances Ras^V*12*^ Hyperplastic Phenotype

We and others have previously demonstrated that the ectopic expression of *Ras^V^*^12^ in the dorsal domain of wing discs, by means of an *apterous-*Gal4 line (*Ap>GFP*; *Ras^V^*^12^), caused tissue overgrowth and the formation of ectopic folds [24,25] (Figure 1B,B’,F). To analyse the role of integrins in *Ras^V^*^12^-mediated oncogenesis, we decided to reduce integrin levels in tumour cells by expressing a *mys*-specific RNAi, a method that was shown to cause a strong reduction in βPS protein levels in wing discs [14]; Appendix A). In agreement with previous results, we found that the expression of a *mys* RNAi in the dorsal compartment of wing discs (*Ap>GFP*; *mys^RNAi^*) induced the formation of ectopic folds and a reduction in its size (Figure 1C,C’,F; [14]). In addition, we found that the downregulation of the βPS integrin function in *Ras^V^*^12^ cells (*Ap>GFP*; *mys^RNAi^*; *Ras^V^*^12^) enhanced the formation of folds and overgrowth due to *Ras^V^*^12^ overexpression alone (Figure 1D,D’,F). We have previously shown that reducing integrin levels in the central region of the wing disc induced capase-dependent cell death [14]. In addition, we found that integrins promoted cell survival by inhibiting the expression of the pro-apoptotic gene *hid*, as reducing *hid* levels rescued apoptosis due to integrin knockdown [14]. Furthermore, we demonstrated that the increase in apoptosis due to the reduction in integrin levels was rescued by the ectopic expression of *Ras^V^*^12^ [14]. In this context, the increase in the fold formation and overgrowth observed in *Ras^V^*^12^ wing discs when reducing integrin levels could be just due to the ability of *Ras^V^*^12^ to rescue the apoptosis of *mys^RNAi^* cells. To test this possibility, we analysed the effects of directly inhibiting apoptosis in wing discs carrying integrin mutant cells in fold formation. To do this, we expressed a *hid*-specific RNAi in *ap>*GFP; *mys^RNAi^* wing discs (*Ap>GFP*; *mys^RNAi^*; *hid^RNAi^*). We found that, although the overexpression of a *hid^RNAi^* in *Ap>GFP*; *mys^RNAi^*wing discs led to an increase in fold formation and overgrowth (Figure 1E,E’,F), the increase was smaller than that found when overexpressing *Ras^V^*^12^ (Figure 1D,D’). Altogether, these results suggest that the ability of *Ras^V^*^12^ overexpression to induce the overgrowth of integrin mutant tissues goes beyond its capacity to rescue the survival of integrin mutant cells.

The expression of β1 integrins in cancer cells is altered, being up- or downregulated depending on the tumour type, stage and microenvironment, an issue that remains controversial [2]. Here, we found that the levels of βPS integrin in *Ras^V^*^12^ wing disc epithelial cells were significantly higher than those found in wild-type neighbouring cells (Appendix A). As expected, the expression of a *mys^RNAi^* in these tumour cells led to a reduction in βPS levels (Appendix A).

Even though the ectopic expression of *Ras^V^*^12^ in wing disc cells alone does not affect cell polarity ([26]; Figure 2A–C’), the loss of polarity genes enhances the *Ras^V^*^12^ hyperplastic phenotype [27]. In addition, there is cumulative evidence that integrin-mediated polarity is implicated in cancer (reviewed in [28]). Furthermore, integrin-mediated cell polarity restrains oncogenesis in the fly midgut [29]. Thus, we analysed whether cell polarity was affected in *Ras^V^*^12^; *mys^RNAi^* disc cells. First, we analysed whether cell polarity was perturbed in the absence of integrins in wing epithelial disc cells and found it was not (Figure 2D,D’). In addition, we found that the downregulation of integrins did not alter the polarity of *Ras^V^*^12^ cells (Figure 2E,E’).

Similarly, while the expression of *Ras^V^*^12^ on its own does not grossly affect the BM, loss of cell polarity in these tumour cells induces BM degradation and the formation of aggressive, malignant tumours (reviewed in [30]). In addition, recent studies have proven that integrins present in tumour cells, or tumour-associated cells, are involved in the remodelling of the ECM (reviewed in [31]). Defects in the assembly of the main BM components, Laminins or Col IV result in failure in the accumulation of the BM component Perlecan [32]. Thus, to test whether the BM was affected in our tumour model, we decided to analyse the levels and localization of Perlecan. We found that the elimination of integrins did not affect the deposition of the BM in either normal or tumour cells (Figure 2F–I’).

### 3.2. Integrin Knockdown Enhances Ras^V*12*^-Associated Cell Shape Changes and Growth

The formation of additional folds could be due to increased cell proliferation, changes in cell shape, cell growth or a combination of these factors. Staining with an antibody against phosphorylated Histone H3 (PH3) showed that, in agreement with previous results [24,25,33,34], the overexpression of *Ras^V^*^12^ resulted in a reduction in the number of PH3^+^ cells (Appendix A). This was not affected by the downregulation of integrin expression (Appendix A). To analyse possible cell shape changes, cell plasma membranes were visualized using the F-actin marker Rhodamine-Phalloidin (RhPh) (Figure 3). We found that, in agreement with previous results, *Ras^V^*^12^ and integrin mutant expressing cells underwent a columnar to cuboidal cell shape change, showing reduced height and increased apical and basal surfaces compared to control cells (Figure 3A–C’’’,F–H) [18,25]. Furthermore, we found that the elimination of integrins exacerbated the cell shape change of *Ras^V^*^12^ cells (Figure 3D–D’’’,F–H). Again, this was not just due to the ability of *Ras^V^*^12^ to rescue apoptosis of integrin mutant cells, as inhibiting apoptosis in *mys^RNAi^* cells does not cause the same shape changes as overexpressing *Ras^V^*^12^ (Figure 3E–E’’’,F–H).

Prolonged expression of *Ras^V^*^12^ promotes G1-S phase transition, as a consequence of E2F activation and cyclin E induction [24]. The use of Fly-Fucci, which relies on fluorochrome-tagged degrons from cyclin B -degraded during mitosis- and E2F1 -degraded at the onset of S phase-, verified the role of *Ras^V^*^12^ in inducing transition from G1 to S phase and also showed that *Ras^V^*^12^ expressing tissues exhibited a reduction of the number of cells in S phase and an accumulation in G2, compared to control tissues [34]. These results are consistent with the previously reported role of *Ras^V^*^12^ in promoting cellular growth and senescence in *Drosophila* imaginal epithelia [24,35]. Using Fly-Fucci, we found that, similar to *Ras^V^*^12^ overexpression [34], Figure 4B–C’), downregulation of integrin levels resulted in an increase in the number of cells in G2 (Figure 4D,D’), an increase that did not depend on cell death (Figure 4F,F’). In addition, reducing integrin levels increased the number of *Ras^V^*^12^ tumour cells in G2 (Figure 4E,E’).

Overall, these results suggest that integrins can restrain *Ras^V^*^12^-mediated tissue hyperplasia via the modulation of cell shape changes and cellular growth. Given the limitations enforced by the peripodial membrane, the detected cell shape changes and increase in cellular growth size observed in all experimental conditions could explain the formation of extra folds.

### 3.3. Integrins Restrict the Ability of Ras^V*12*^ Tumour Cells to Stimulate Apoptosis in Neighbouring Wild-Type Cells

Even though the overexpression of *Ras^V^*^12^ can promote cell survival, several studies have shown that it can also stimulate the death of nearby wild-type cells [36,37]. Accordingly, in a recent study, we detected a clear enrichment in the levels of the apoptotic marker cleaved Dcp1 in wild-type (GFP negative) ventral cells located at the D/V boundary in *Ap>GFP*; *Ras^V^*^12^ discs [25]; Appendix A). In addition, as mentioned above, we have previously shown that reducing integrin levels in the whole wing pouch induces caspase-dependent cell death, which is suppressed by oncogenic Ras [14]. Here, we found that although apoptosis was reduced in the dorsal region of *Ap>GFP*; *Ras^V^*^12^; *mys^RNAi^* (Appendix A), compared to *Ap>GFP*; *mys^RNAi^* discs (Appendix A), in agreement with our previous results [14], there was an increase in Dcp1 levels at the D/V boundary, similar to what happens in *Ap>GFP*; *Ras^V^*^12^ discs (Appendix A). Furthermore, the elimination of integrins increased the ability of *Ras^V^*^12^ cells to induce apoptosis in neighbouring wild-type cells, as measured by Dcp1 levels (Appendix A).

The capacity of *Ras^V^*^12^ cells to induce apoptosis of neighbouring wild-type cells has been shown to be mediated by activation of the JNK pathway [36,37]. Accordingly, in a recent study, using an antibody against phosphorylated JNK (pJNK), we detected a clear enrichment of JNK activity in wild-type (GFP negative) ventral cells located at the D/V boundary in *Ap>GFP*; *Ras^V^*^12^ discs [25], Figure 5A–B’,F). We have also shown that JNK mediates apoptosis induced by loss of integrin function [14]. Here, we found that although JNK activity was reduced in the dorsal region of *Ap>GFP*; *Ras^V^*^12^; *mys^RNAi^* (Figure 5D,D’,F), compared to *Ap>GFP*; *mys^RNAi^* discs (Figure 5C,C’,F), pJNK levels at the D/V boundary were higher than those found in *Ap>GFP*; *Ras^V^*^12^ discs (Figure 5B,B’,F).

Induction of apoptosis and activation of the JNK pathway in wild-type neighbours of *Ras^V^*^12^ cells have been partially explained by an increase in tissue compaction due to the enhanced growth of tumour cells [38]. Thus, the rise in apoptosis (Appendix A) and JNK activity (Figure 5) we observed in wild-type cells located at the D/V boundary in *Ap>GFP*; *Ras^V^*^12^; *mys^RNAi^*, compared to *Ap>GFP*; *Ras^V^*^12^ discs, could be due to the increase in the formation of folds in the tumorigenic tissue upon integrin removal.

### 3.4. Integrins Limit the Ability of Epithelial Ras^V*12*^ Tumour Cells to Leave the Epithelium and Move

We recently found that a proportion of wing discs expressing GFP and *Ras^V^*^12^ in the dorsal domain showed GFP^+^
*Ras^V^*^12^ cells in the ventral domain, some of which were positive for the apoptotic marker cleaved Dcp1 [25]; Figure 6A,B,F). This result led us to suggest that the expression of *Ras^V^*^12^ in wing disc cells could induce an invasive behaviour. Here, using the Dcp1 marker to distinguish between dead and alive cells, we found that 39.3% of *Ap>GFP*; *Ras^V^*^12^ wing discs presented alive GFP+ cells in the ventral domain (Figure 6B–B’’’,F), in contrast to the 8.2% found in control wing discs (Figure 6A–A’’’,F). In addition, alive ventral GFP+ cells were only seen in 14.3% of wing discs with reduced integrin levels (Figure 6C–C’’’,F). However, here, we found that reducing integrin function increased up to 63.3% the percentage of *Ras^V^*^12^ wing discs showing invading GFP^+^ cells (Figure 6D–D’’’,F). Once more, this was not just due to the ability of *Ras^V^*^12^ to rescue the death of integrin mutant cells, as preventing apoptosis in *mys^RNAi^* cells only increased the penetrance of the invasive phenotype up to 20.6% (Figure 6E–E’’’,F).

Our previous analysis had shown that eliminating integrin function in *Ras^V^*^12^ cells promoted their basal extrusion [14,25]. To study in more detail the behaviour of these tumour cells, we dissected, cultured and monitored lively control and experimental third instar wing discs expressing GFP in the dorsal domain (see Materials and Methods). Live imaging analysis of confocal sections of cultured wing discs allowed us to watch the behaviour of dorsal GFP^+^ cells for up to 180 min (Figure 7).

Analysis of the control *Ap>*GFP wing discs showed that dorsal cells stayed confined within their compartment throughout the incubation period (Figure 7B–B’’’, Appendix A). Conversely, a few GFP^+^ cells from *Ap>*GFP; *Ras^V^*^12^ wing discs were observed leaving the dorsal compartment and moving into the GFP^−^ ventral compartment (Figure 7C–C’’’, Appendix A). Some of these cells looked smaller and rounder than control cells (arrowheads in 7C–C’’’) and we assumed these were dying cells. This was supported by live imaging analysis of *Ap>*GFP; *mys^RNAi^* wing discs, showing that cells moving out of the ventral domain were also small and round (Figure 7D–D’’’, Appendix A). In contrast, we found that most *mys^RNAi^*; *Ras^V^*^12^ wing disc cells looked normal and were able to move into the ventral domain further than *Ras^V^*^12^ cells (Figure 7E–E’’’, Appendix A). This was not just a consequence of the ability of *Ras^V^*^12^ to rescue the death of integrin mutant cells, as only a few small GFP^+^ cells were found moving into the ventral compartment in *mys^RNAi^*; *hid^RNAi^* wing discs (Figure 7F–F’’’, Appendix A).

Together, the data demonstrate that integrins can restrict the ability of *Ras^V^*^12^ tumour cells to invade adjacent tissues.

## 4. Discussion

Integrins have been implicated in nearly every step of cancer progression, including initiation, proliferation, survival and metastasis. However, accumulating evidence has shown that integrins can act not only as tumour promoters but also as tumour suppressors (reviewed in [2]). This is especially the case for integrins that mediate cell adhesion to laminins, major components of the basement membranes surrounding most organs and tissues [39]. The ability of integrins to promote or suppress oncogenesis depends, among other factors, on the cell type, the oncogene context and the stage of the tumour, (reviewed in [40]). Here, we find that, in the *Drosophila* wing disc epithelium, the βPS containing integrins act as suppressors of tumours induced by a gain of function of the oncogenic form of *Ras*, *Ras^V^*^12^. We show that while the depletion of βPS integrins does not affect the polarity and proliferation of *Ras^V^*^12^ tumour cells, it increases their growth and invasive behaviour. In addition, we show that the elimination of integrins enhances the ability of *Ras^V^*^12^ tumour cells to affect the tumour microenvironment.

To better understand the genetic bases and the development of cancer, researchers have used small animal models, such as mice, zebrafish and *Drosophila* [41]. In particular, mice are the most widely used to analyse the role of integrins in tumour development. The use of conditional genetic mouse models has pointed to opposing roles for the *β*_1_ family of integrins as tumour enhancers or suppressors. Deletion of *β*_1_ integrins blocks and delays cancer initiation in the Polyoma middle T (PyMT) and ErbB2 oncogene models, respectively [5,42]. Likewise, in the RIPTag model for pancreatic cancer and in squamous cell carcinomas, the deletion of *β*_1_ integrins leads to impaired tumour growth and metastasis [43,44]. In contrast, in other cancer models, *β*_1_ integrins appear to play a tumour suppressor-like role. Thus, the deletion of *β*_1_ integrins in the TRAMP prostate adenocarcinoma model led to an increase in the expansion of the tumour cell population, an enhancement of the rate of tumour progression and a reduction in overall animal survival [45]. In contrast to the wide use of mice to study the role of integrins in cancer development, few studies have been reported using the zebrafish or the *Drosophila* models to analyse this issue. The zebrafish has become a promising model for vertebrates in recent years, as it is small, transparent, low cost and reproduces fast. At present, zebrafish xenografts have been used to analyse the role of integrins in metastasis. These studies have shown that disrupting the expression of some integrins, such as *β*_1_, α_3_ or α_6_ in cancer cells reduces their ability to extravasate or metastasize when transplanted in zebrafish, suggesting a role for integrins as tumour promoters in this context [46,47,48]. Intriguingly, with respect to *Drosophila*, despite the fact that it has been extensively used to study cancer, there is little evidence for its use as a platform to analyse the role of integrins in tumour development. Using *Drosophila* Intestinal Stem Cell tumours in the adult midgut, it was shown that integrins were required for the initiation of hyperplasia upon loss of adenomatous polyposis coli and for the growth of tumours induced by the gain of an oncogenic form of the transcription co-factor Yki [49,50]. These results proposed a role for integrins as tumour-promoting factors. In contrast, here, we show that integrins can act as tumour suppressors in the *Drosophila* wing imaginal disc epithelium, suggesting that, the dual role of integrins as tumour modulators is conserved. In the future, *Drosophila* could be used to understand the opposing tumour modulating activities conferred by integrins. This will facilitate further development of drugs targeting integrin signalling pathways.

The role of integrins in tumour growth is also complex. Thus, while loss of the α_2_β_1_ integrin does not affect tumour growth in vivo or in vitro, the α_2_-null/Neu tumour cells demonstrate enhanced anchorage-independent growth [6]. In *Drosophila*, loss of integrin function inhibits the proliferation of Yki-driven intestinal stem cell tumours in the adult [50]. As mentioned above, ectopic expression of *Ras^V^*^12^ in the *Drosophila* wing disc produces hyperplasia, as a consequence of increased cell growth, accelerated G1−S transition and cell shape changes [24]. In agreement with this, we have recently shown that ectopic expression of *Ras^V^*^12^ in wing disc cells induces a cell shape change from columnar to cuboidal [25]. Here, we show that downregulation of integrin expression increases the formation of extra folds due to *Ras^V^*^12^ overexpression, by enhancing cell shape changes and cellular growth. We have previously shown that the removal of integrins on its own induces a change in cell shape similar to that caused by *Ras^V^*^12^ overexpression [18]. In addition, we have recently found that integrins regulate cell shape in the wing disc by modulating Myosin II activity and dynamics [14]. 

Similarly, increased EGFR/Ras signalling has been shown to affect cell shape through the regulation of Myosin II dynamics [51,52]. Thus, the increase in the folding phenotype we observed in *Ras^V^*^12^ wing discs when downregulating integrin expression could just be an additive effect of simultaneously disrupting two pathways independently regulating cell shape. Alternatively, integrins could modulate oncogenic *Ras* signalling. In fact, many crosstalk between integrins and oncogenic *Ras* have been proposed, including the regulation of RTK activity by integrins [53]. However, in most cases, these have been shown to increase *Ras* signalling (reviewed in [54]), rather than limit it, which is what we find here. In the future, it will be interesting to investigate the mechanisms by which integrins restrict oncogenic *Ras* function.

Integrins not only regulate the properties of the cancer cells but also their ability to alter the tumour microenvironment, which includes the extracellular matrix and surrounding cells, stimulating non-autonomous tumour progression (reviewed in [55]. Here, we find that while the elimination of integrins in *Ras^V^*^12^ wing disc cells does not seem to clearly affect the adjacent basement membrane, it increases the ability of the tumour cells to cause activation of the JNK pathway and non-autonomous death in neighbouring wild-type tissue. This effect has been proposed to increase the proliferation of tumour cells through the secretion of growth factors and Eiger (tumour necrosis factor homolog) [36,37]. Here, we would like to propose that this non-autonomous effect could also contribute to tumour invasion by secreting factors that could potentiate the migratory behaviour of tumour cells. The contribution of the non-autonomous effects of cancer cells to tumour invasion is an exciting area for future investigation.

Most studies have shown that the upregulation or overexpression of integrins associates with cancer metastasis, conferring integrins a pro-metastatic activity (reviewed in [40]). Furthermore, integrin inhibitors, such as monoclonal antibodies or RGD peptide analogs, are in clinical trials as metastatic suppressors [56]. However, other studies have reported that integrins can also inhibit metastasis, demonstrating that the role of integrins in cancer metastasis is also complex. For instance, the α_3_β_1_ and α_2_β_1_ integrins have been shown to suppress metastasis of prostate and breast cancer [6,57,58,59]. Moreover, the same integrin can play opposite roles in the progression of different types of cancers. Depletion of the integrin α_9_ reduces tumour metastasis in triple-negative breast cancer [60], while its overexpression suppresses hepatocarcinoma invasion [61]. Here, we find that while the expression of *Ras^V^*^12^ alone is not sufficient to create invasive tumours in wing disc epithelial cells, the elimination of integrins in these tumour cells triggers an invasive behaviour. As integrins are the main receptor implicated in cell migration, how can their elimination induce invasion? Even though collective cell movement of primary melanoma explants requires β1-integrin mediated cell-matrix adhesion, impairment of cluster cohesion resulted in detachment and dissemination of amoeboid single tumour cells [62]. Similarly, when cultured in 3D ECM scaffolds, the blocking β_1_ integrin function swapped the migratory behaviour of human and mouse E-cadherin–positive triple-negative breast cancer (TNBC) cells from an integrin-dependent collective movement to an integrin-independent “amoeboid” crawling and dissemination [7]. We have previously shown that eliminating integrins from *Ras^V^*^12^ wing disc epithelial cells induces their detachment [14]. Here, we show that these tumour cells are able to move and invade nearby regions. Based on these results, we would like to suggest a conserved role for the β_1_ family of integrins as promoters of epithelial phenotype and suppressors of dissemination. Understanding the apparent metastatic suppression activity conferred by integrins in certain cancer contexts is crucial to developing proper treatment strategies targeting integrins.

A variety of *Drosophila* cancer models has been established in several tissues besides the wing imaginal disc, such as the gut, the brain and the eye imaginal disc, by means of transgenesis, genome editing, transplantation and drug-induced toxic damage. These tumour models have provided new concepts and findings in cancer biology that have significant parallels and relevance to human cancer. Our findings establish the *Drosophila* wing imaginal disc as a platform to address the complex role of integrins in cancer. We can now extend these studies to the other *Drosophila* cancer models to further explore the diverse biological functions integrins perform in cancers. As targeting integrins has shown huge potential in the diagnosis and treatment of cancer, deeper mechanistic insights are necessary to better comprehend the mechanisms of the integrin-mediated biological behaviour of cancer cells.

## 5. Conclusions

Using the *Drosophila* wing disc epithelium, we show that the βPS integrin, the orthologue of vertebrate β1, can act as a suppressor of tumours induced by a gain of function of the oncogenic form of *Ras*, *Ras^V^*^12^. We show that the βPS integrin restrains the growth, delamination and invasiveness of *Ras^V^*^12^ tumour cells, as well as their ability to affect the tumour microenvironment. These results strongly suggest that the role of the integrin of the β1 family as tumour suppressors might be evolutionarily conserved. In the future, it will be interesting to study the contribution of the βPS integrin to other *Drosophila* cancer models to further explore the multiple biological functions mediated by integrins in cancers. *Drosophila* can be used to understand the mechanisms underlying the complex tumour modulating activities conferred by integrins, thus facilitating the design of cancer drugs targeting integrins.

## Figures and Tables

**Figure 1 cancers-15-05432-f001:**
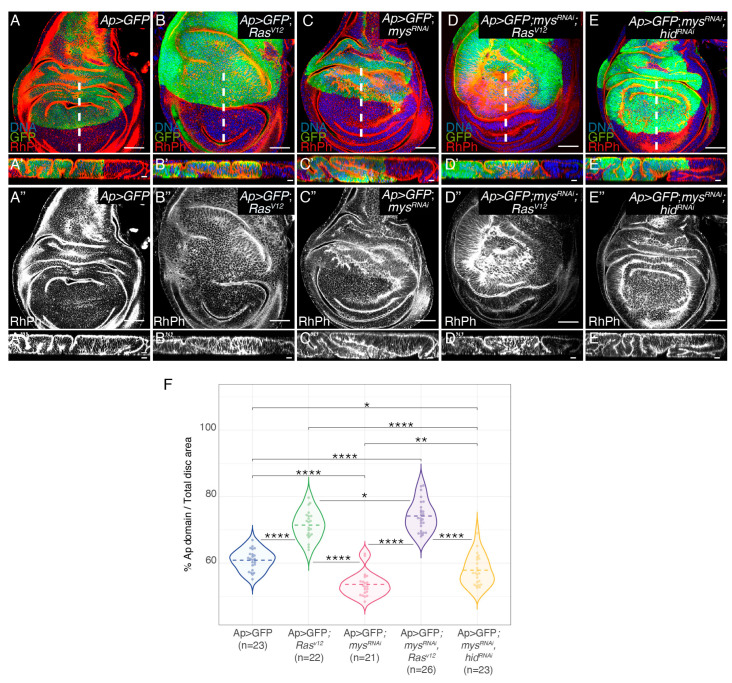
**Integrin knockdown enhances *Ras^V*12*^* hyperplasia in the *Drosophila* wing disc.** (**A**–**E**,**A’’**–**E’’**) Confocal views of third-instar larvae wing discs of the indicated genotypes, stained with anti-GFP (green), Rhodamine Phalloidin (RhPh) to detect F-actin (red in **A**–**E’**, white in **A”**–**E”’**) and Hoechst for DNA detection (blue). (**A’**–**E’**,**A’’’**–**E’’’**) Confocal yz sections of wing discs, along the white dotted line shown in (**A**–**E**). (**F**) Violin plots of the percentage of the apterous domain area versus total area per disc. The statistical significance of differences was assessed with a Welch test, ****, **, and * *p* values <0.0001, <0.01 and <0,05, respectively. Scale bars 50 μm (**A**–**E**,**A’’**–**E’’**) and 10 μm (**A’**–**E’**,**A’’’**–**E’’’**).

**Figure 2 cancers-15-05432-f002:**
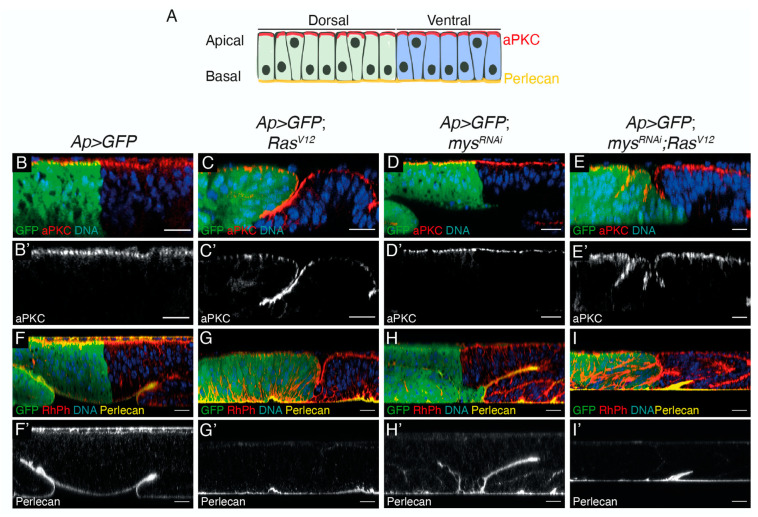
**Inhibition of integrin expression in *Ras^V*12*^* tumoural cells does not affect cell polarity or BM deposition.** (**A**) Scheme of apico-basal polarity in the pseudostratified epithelium of the *Drosophila* wing disc, showing the Dorsal (green) and Ventral (blue) domains, the localization of the apical determinant aPKC (red) and the BM component Perlecan (yellow). (**B**–**I’**) Confocal xz sections of third-instar larvae wing discs of the designated genotypes, stained with anti-GFP (green), Hoechst (DNA, blue), aPKC (red in **B**–**E**, white in **B’**–**E’**) and Perlecan (yellow in **F**–**I**, white in **F’**–**I’**). Scale bars 50 μm (**B**–**I’**).

**Figure 3 cancers-15-05432-f003:**
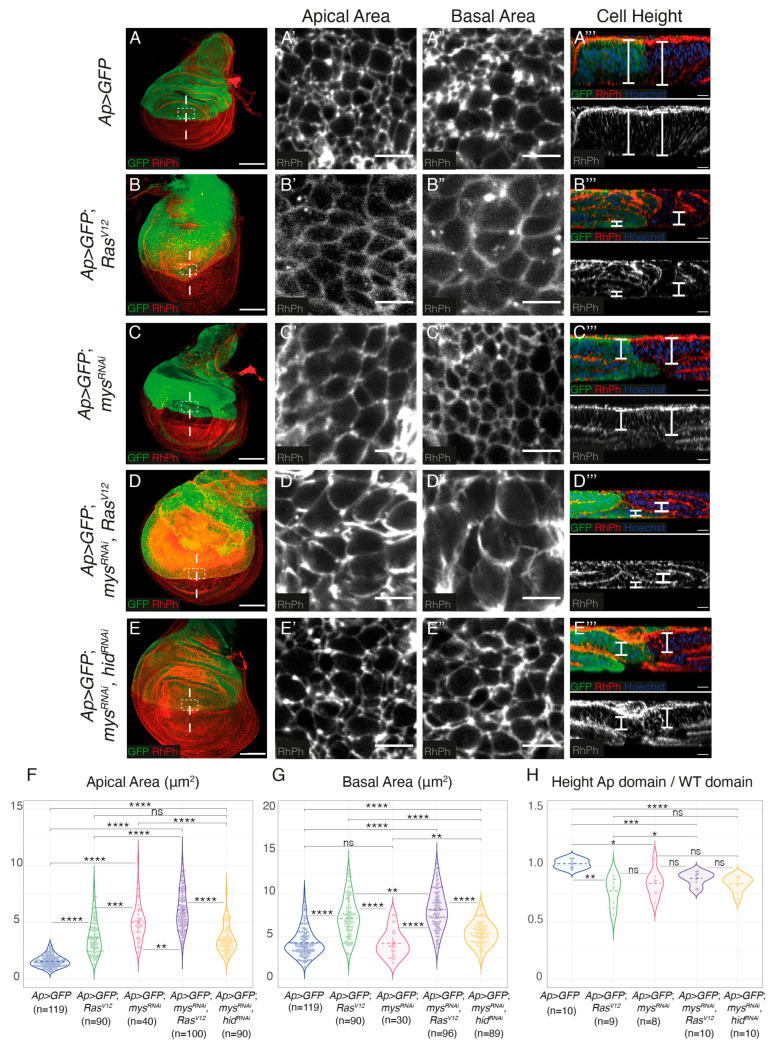
**Integrin expression downregulation increases *Ras^V^*^12^-dependent cell shape changes and growth in *Drosophila* wing discs**. (**A**–**E**) Maximal projection of confocal images of third-instar larvae wing discs of the designated genotypes, stained with anti-GFP (green), RhPh (red in **A**–**E**, white in **A’**–**E’’**) and Hoechst for DNA detection (blue, **A’’’**–**E’’’**). (**A’**–**E’**) Apical and (**A”**–**E”**) basal surface views of the region specified in the white boxes in (**A**–**E**). (**A’’’**–**E’’’**) Confocal *xz* sections along the white dotted lines of wing discs shown in (**A**–**F**). The apical side of wing discs is at the top. White brackets indicate cell height. (**F**–**H**) Violin plots of the apical cell area (**F**), basal cell area (**G**) and cell height (**H**) of the indicated genotypes. The statistical significance of differences was measured using the welch-test, ****, ***, **, * and ‘ns’ *p* values <0.0001, <0.001, <0.01, <0.05 and >0.05, respectively. Scale bars 50 μm (**A**–**E**), 5 μm (**A’**–**E’**,**A”**–**E”**) and 10 μm (**A’’’**–**E’’’**).

**Figure 4 cancers-15-05432-f004:**
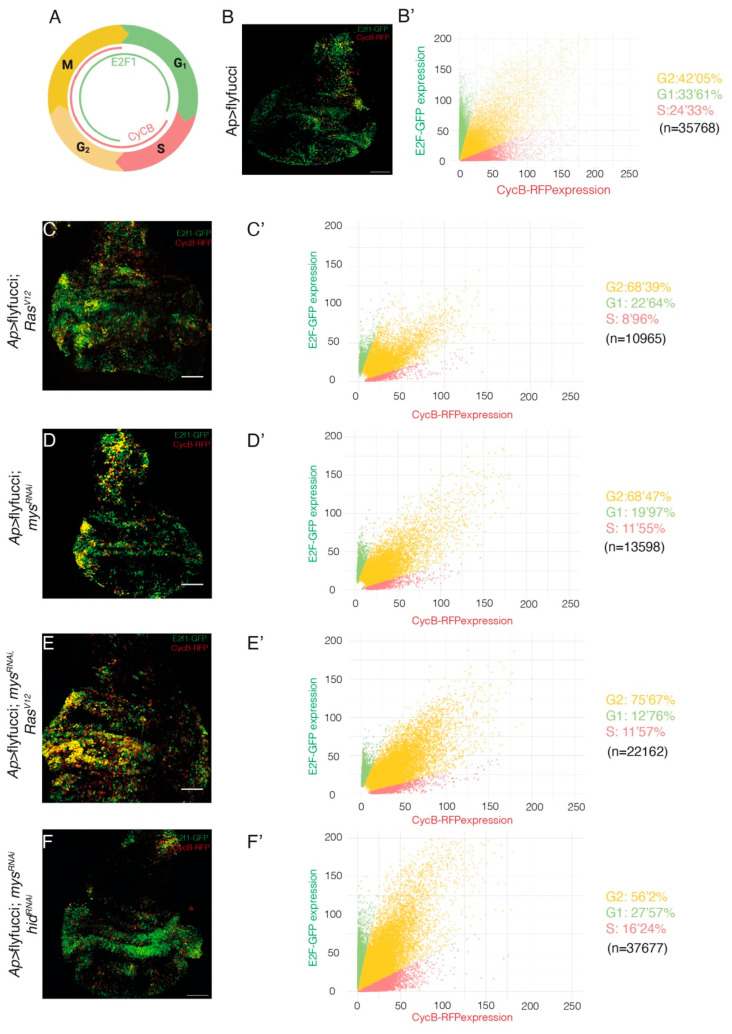
**Integrin knockdown enhances the changes in cell cycle progression in *Drosophila Ras^V^*^12^ wing disc cells**. (**A**) Scheme depicting CycB-RFP and E2F1-GFP nuclear expression during the cell cycle. (**B**–**F**) Maximal projection of confocal images of the apterous domains of third-instar larvae wing discs of the indicated genotypes, expressing E2f1-GFP (green) and CycB-RFP (red). (**B’**–**F’**) Scatter plots representing the fluorescence intensity of E2f1-GFP (green) and CycB-RFP (red). The statistical significance of differences was measured using the Xi-square test, with all comparisons having a *p*-value < 0.0001. Scale bars 50 μm (**B**–**F**).

**Figure 5 cancers-15-05432-f005:**
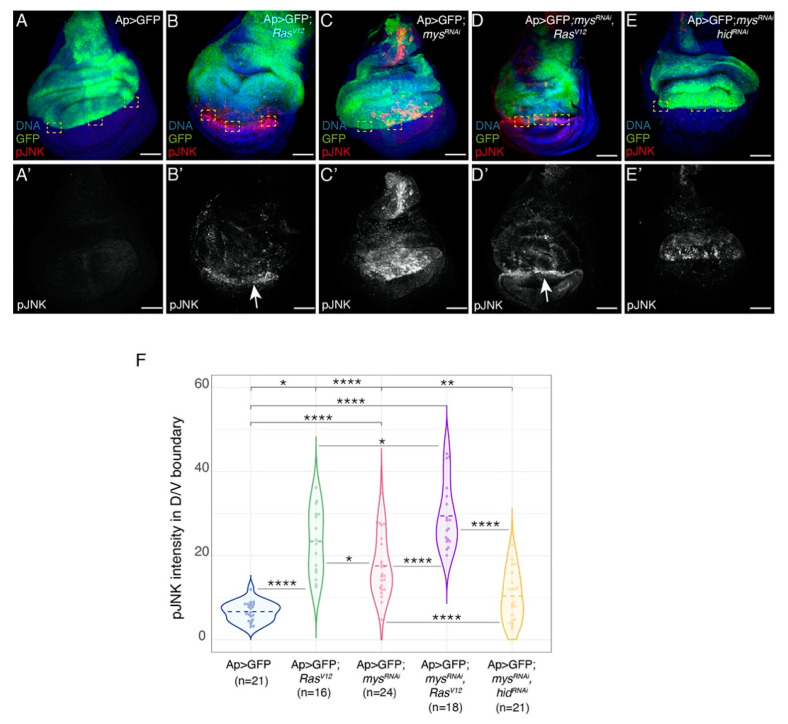
**Integrin knockdown increases the ability of *Ras^V^*^12^ to induce JNK activity in wild-type adjacent cells.** (**A**–**E’**) Maximal projection of confocal views of third-instar wing discs of the specified genotypes, stained with anti-GFP (green), anti-pJNK (red in **A**–**E**, white in **A’**–**E’**) and Hoechst (DNA, blue). (**F**) Violin plots of mean fluorescent pJNK intensity at the dorsal-ventral boundary of wing discs of the designated genotypes. The statistical significance of differences was assessed with the welch-test, ****, ** and * *p* values < 0.0001, <0.01 and <0.05, respectively. Scale bars, 50 μm (**A**–**E’**).

**Figure 6 cancers-15-05432-f006:**
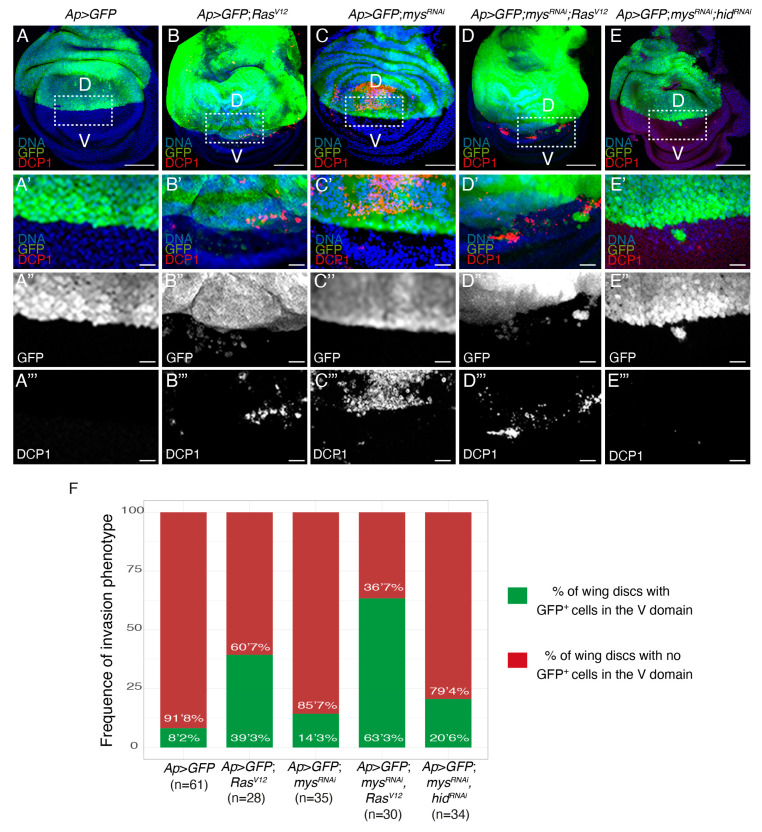
**Integrins downregulation enhances the invasive behaviour of *Ras^V^*^12^ cells.** (**A**–**E**) Confocal views of third-instar larvae wing discs of the designated genotypes, stained with anti-GFP (green in **A**–**E’** and white in **A’’**–**E’’**), DCP1 (red in **A**–**E’** and white in **A’’’**–**E’’’**) and Hoechst (DNA, blue) (**A**–**E’**). (**A’**–**E’’’**) Magnifications of the white boxes in (**A**–**E**), respectively. (**F**) Bar plot of the frequency of appearance of non-dead GFP^+^ cells in the ventral domain. Scale bars 50 μm (**A**–**E**) and 10 μm (**A’**–**E’’’**).

**Figure 7 cancers-15-05432-f007:**
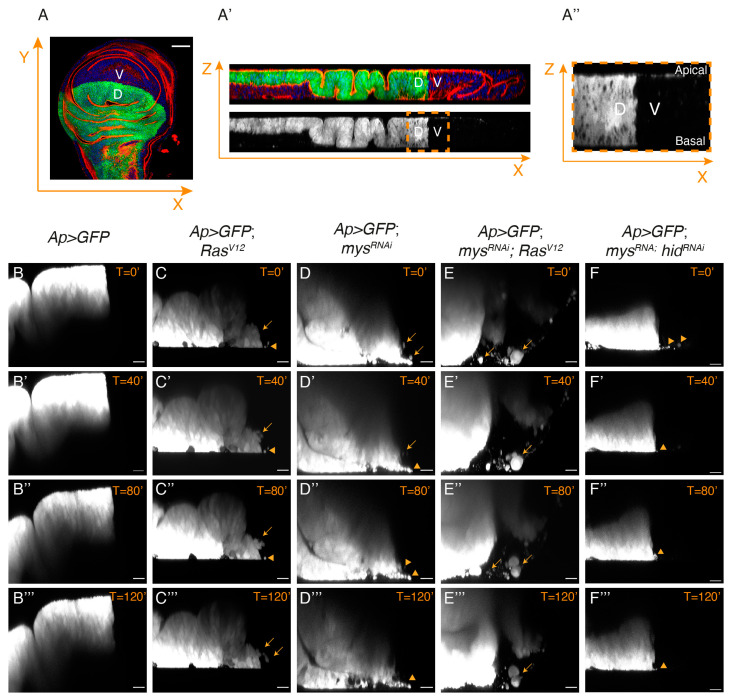
**Integrins regulate the invasive capacity of *Ras^V^*^12^ cells**. (**A**–**A’’**) Scheme illustrating the region of the wing disc where movies had been taken. (**A**) Confocal view of a third-instar larvae wing disc, stained with Hoechst (blue), RhPh (red) and anti-GFP (green), and (**A’**) and its corresponding orthogonal view. (**A’’**) Zoom-in of the region in the yellow dotted box in (**A’**), corresponding to the region where movies had been taken. (**B**–**F’’’**) Confocal images taken with a 40 min difference of live wing discs of the designated genotypes.

## Data Availability

Data is contained within the article or Appendix A.

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
