# Peer review of "Integrins Can Act as Suppressors of Ras-Mediated Oncogenesis in the Drosophila Wing Disc Epithelium"

_cancers, 2023, doi:10.3390/cancers15225432_

Round 1

Reviewer 1 Report

Comments and Suggestions for Authors

The manuscript of Millán and Martin-Bermudo is a well-designed and executed study addressing an important topic in current cancer research, which is the cross-talk between cancer cells and their microenvironment. The authors use the power of Drosophila genetics to investigate how integrins cooperate with the oncogenic form of RasRasV12 in the wing disc epithelium, a great model to characterize features of epithelial growth, delamination and invasive behaviour.

The authors of the manuscript have done terrific work putting in the appropriate context and providing significant information that strengthens the notion that integrins can act as a suppressor of tumours induced by a gain-of-function mutation of the oncogenic form of RasRasV12. Given that their data derived from an in vivo model, adds further importance to their conclusions.

I only have one main point that the authors need to address. The authors conclude that “elimination of integrins in RasV12 wing disc cells does not seem to clearly affect the adjacement basement membrane”, as shown in Sup Fig. 2F-I’’, using Perlecan as a marker. I suggest to check whether the deposition and accumulation pattern of other ECM components (e.g. Laminin) similarly remain unaffected. By examining additional markers the authors will corroborate their conclusion.

I also suggest that Sup. Fig 1 and 2 should be part of the main manuscript because they contain crucial data.

Author Response

The manuscript of Millán and Martin-Bermudo is a well-designed and executed study addressing an important topic in current cancer research, which is the cross-talk between cancer cells and their microenvironment. The authors use the power of Drosophila genetics to investigate how integrins cooperate with the oncogenic form of RasRasV12 in the wing disc epithelium, a great model to characterize features of epithelial growth, delamination and invasive behaviour.

The authors of the manuscript have done terrific work putting in the appropriate context and providing significant information that strengthens the notion that integrins can act as a suppressor of tumours induced by a gain-of-function mutation of the oncogenic form of RasRasV12. Given that their data derived from an in vivo model, adds further importance to their conclusions.

I only have one main point that the authors need to address. The authors conclude that “elimination of integrins in RasV12 wing disc cells does not seem to clearly affect the adjacement basement membrane”, as shown in Sup Fig. 2F-I’’, using Perlecan as a marker. I suggest to check whether the deposition and accumulation pattern of other ECM components (e.g. Laminin) similarly remain unaffected. By examining additional markers the authors will corroborate their conclusion.

This is a good suggestion. However, unfortunately, there are not antibodies available for Laminins or Col IV. This implies that in order to visualize any of these BM components, we will need to generate flies that, besides carrying the constructs necessary to simultaneously overexpress RasV12 and downregulate integrin function in the wing imaginal disc, UAS-mysRNAi, UAS; RasV12 and ApGal4, they will need to also carry available GFP insertion in the genes encoding for these BM components. This will require to generate recombinant flies to accommodate all these constructs, and, unfortunately, we are not able to perform these experiments due to lack of revision time. However, I would like to add that we chose to look at Perlecan on purpose, as, in fact, it is a readout of BM assembly, since it has been shown that Drosophila Laminins and Col IV are required for the incorporation of Perlecan into the BMs of all tissues (Matsubayashi et al. 2017, Current Biol.). In this context, normal accumulation of Perlecan reflects normal deposition and assembly of the main BM components. We have incorporated this rationale in the main text.

I also suggest that Sup. Fig 1 and 2 should be part of the main manuscript because they contain crucial data.

We have changed Sup. Fig.2 to main Fig.2. We have not done the same with Sup Fig.1, as we do not investigate further why the levels of integrins are slightly upregulated when overexpressing RasV12, because our work mainly focuses on the effects of reducing integrin expression in RasV12 cells.

Reviewer 2 Report

Comments and Suggestions for Authors

In the manuscript by Millán et al., the authors present findings demonstrating the inhibitory role of integrins in Ras-mediated oncogenesis. They observe that reducing the function of βPS integrin enhances the hyperplasia phenotype and invasive behavior induced by activated Ras (RasV12) in the Drosophila wing disc. While investigating the underlying mechanism, the authors provide evidence that knockdown of βPS integrin increases RasV12-dependent cell shape changes, cell cycle progression, and the ability of RasV12 to induce apoptosis in neighboring wild-type cells. Lastly, the authors demonstrate that the elimination of integrin function in RasV12 cells promotes their basal extrusion.

Overall, the observed phenotypes are noteworthy, and the identification of βPS integrin as a tumor suppressor of Ras-mediated tumorigenesis in Drosophila is interesting. Nonetheless, the mechanistic experiments are deficient, with some of them possibly having alternative explanations. To ensure the credibility of the research, several concerns need to be thoroughly addressed before consideration for publication.

1.     Figs 1. The authors show that integrin knockdown enhances RasV12 hyperplasia in the Drosophila wing disc. However, the authors only show that ectopic expression of RasV12 or reducing integrin by expressing mys RNAi in the dorsal compartment of wing discs by means of an apterous-Gal4 line.

1)    Quantification of experiments is inconsistent. Figs 1 lack quantification and statistical analysis. The phenotype of ectopic folds is confusing and difficult to obtain the conclusion that downregulation of βPS integrin enhances the formation of folds due to Ras activation.

2)    In Drosophila, GFP-marked mosaic clones of cells expressing RasV12 overgrow to develop into tumors. The overgrowth phenotype can be readily ascertained by examining clone size in dissected eye-antenna or wing imaginal discs. They could induce myc (mys1, null mutations of integrin genes ,BDSC: 23862 Genotype: mys1 P{neoFRT}19A/FM7c) or RasV12 single mutant clones or RasV12mys double mutant clones and examined the hyperplasia phenotype of these mutant clones in similarly aged third instar eye-antenna or wing discs by using eyFLP or UbxFLP.

2.     Although the authors could address these findings that βPS integrin knock down increases RasV12-dependent cell shape changes, cell cycle progression and the ability of RasV12 to induce apoptosis in neighboring WT cells easily by a simple RNAi experiment. They could test if supplying more integrins in the epithelium promotes attachment to the ECM, by expressing UAS-mys (BDSC:68158) and UAS-βInt-ν, individually or in combination, which inhibit Ras-mediated oncogenesis.

3.     Among these cancer models, the cooperation between RasV12 and the loss of the cell polarity tumor suppressor such scribble, lethal giant larvae (lgl), dlg has been the most extensively investigated. To investigate whether βPS integrin could also have a tumour-suppressive effect in a more aggressive model of RasV12-related tumourigenesis,

1)     they could overactive βPS integrin function in the eye-antennal or wing epithelium in which RasV12 was expressed together with knockdown of the cell polarity gene.

2)     The authors show that integrins can restrict the ability of RasV12 tumor cells to invade adjacent tissues. They could test if loss of mys (mys1, null mutations) can enhance the invasive capacity of tumor cell in the eyFLP Act»GAL4 RasV12 dlgRNAi model by examining the grades of cell invasion into the ventral nerve cord.

4.     In Fig 5F, the green should mean % of wing discs with GFP+ cells in the V domain.

5.     The legends are also too short and unfriendly for non-fly readers. The authors should modify all the corresponding legends.

Author Response

In the manuscript by Millán et al., the authors present findings demonstrating the inhibitory role of integrins in Ras-mediated oncogenesis. They observe that reducing the function of βPS integrin enhances the hyperplasia phenotype and invasive behavior induced by activated Ras (RasV12) in the Drosophila wing disc. While investigating the underlying mechanism, the authors provide evidence that knockdown of βPS integrin increases RasV12-dependent cell shape changes, cell cycle progression, and the ability of RasV12 to induce apoptosis in neighboring wild-type cells. Lastly, the authors demonstrate that the elimination of integrin function in RasV12 cells promotes their basal extrusion.

Overall, the observed phenotypes are noteworthy, and the identification of βPS integrin as a tumor suppressor of Ras-mediated tumorigenesis in Drosophila is interesting. Nonetheless, the mechanistic experiments are deficient, with some of them possibly having alternative explanations. To ensure the credibility of the research, several concerns need to be thoroughly addressed before consideration for publication.

  1. Figs 1. The authors show that integrin knockdown enhances RasV12 hyperplasia in the Drosophila wing disc. However, the authors only show that ectopic expression of RasV12 or reducing integrin by expressing mys RNAi in the dorsal compartment of wing discs by means of an apterous-Gal4 line.

1)    Quantification of experiments is inconsistent. Figs 1 lack quantification and statistical analysis. The phenotype of ectopic folds is confusing and difficult to obtain the conclusion that downregulation of βPS integrin enhances the formation of folds due to Ras activation.

2)    In Drosophila, GFP-marked mosaic clones of cells expressing RasV12 overgrow to develop into tumors. The overgrowth phenotype can be readily ascertained by examining clone size in dissected eye-antenna or wing imaginal discs. They could induce myc (mys1, null mutations of integrin genes ,BDSC: 23862 Genotype: mys1 P{neoFRT}19A/FM7c) or RasV12 single mutant clones or RasV12mys double mutant clones and examined the hyperplasia phenotype of these mutant clones in similarly aged third instar eye-antenna or wing discs by using eyFLP or UbxFLP.

We have now measured the area of the dorsal domain relative to the whole wing disc area in control and experimental wing discs. Our results show that, in agreement with previous results, expression of RasV12 in the dorsal domain induces its overgrowth, which is enhanced by elimination of integrin function. This data has been incorporated into a new Fig.1 (Fig.1F).

  1. Although the authors could address these findings that βPS integrin knock down increases RasV12-dependent cell shape changes, cell cycle progression and the ability of RasV12 to induce apoptosis in neighboring WT cells easily by a simple RNAi experiment. They could test if supplying more integrins in the epithelium promotes attachment to the ECM, by expressing UAS-mys(BDSC:68158) and UAS-βInt-ν, individually or in combination, which inhibit Ras-mediated oncogenesis.

This is an experiment that we tried. However, we found that supplying more integrins, by means of expressing UAS-mys, caused a dominant negative phenotype in the wing disc. This was also shown to be the case when expressing UAS-mys in other tissues, including the embryonic muscles and the gut, and it was explained by the ability of the bPS subunit to form homodimers when in excess. These homodimers cannot bind the ECM ligands but compete with the endogenous integrin heterodimers for intracellular partners, thus resulting in a dominant negative phenotype. We tried to overcome this problem by co-expressing both an a and the b subunit simultaneously, but this also led to a phenotype on its own. These results suggest that integrin levels need to be tightly regulated to allow proper wing disc morphogenesis, but, unfortunately, also precluded us from reaching accurate conclusions from results derived from these experiments.

  1. Among these cancer models, the cooperation between RasV12 and the loss of the cell polarity tumor suppressor such scribble, lethal giant larvae (lgl), dlg has been the most extensively investigated. To investigate whether βPS integrin could also have a tumour-suppressive effect in a more aggressive model of RasV12-related tumourigenesis,

1)     they could overactive βPS integrin function in the eye-antennal or wing epithelium in which RasV12 was expressed together with knockdown of the cell polarity gene.

We have performed this experiment with the aim to further support our hypothesis for a role of integrins as suppressors of RasV12 tumors. In order to do this, we co-expressed in RasV12 wing discs the bPS subunit with an activated form of an alfa subunit carrying a deletion of its cytoplasmic domain, which has been previously shown to behave as an active integrin (Martin-Bermudo et al. 1998). Unfortunately, the expression of the activated integrins on its own on the wing disc already produced a dramatic change in cell shape. We believe expressing this active integrin in RasV12 wing discs with knockdown of the cell polarity gene lgl, might be, not only difficult to achieve, due to the complication of generating flies carrying all the necessary constructs, but, overall, very difficult to interpret.

2)     The authors show that integrins can restrict the ability of RasV12 tumor cells to invade adjacent tissues. They could test if loss of mys (mys1, null mutations) can enhance the invasive capacity of tumor cell in the eyFLP Act»GAL4 RasV12 dlgRNAi model by examining the grades of cell invasion into the ventral nerve cord.

This is an interesting suggestion. As mentioned in the last paragraph of the discussion “A variety of Drosophila cancer models has been established in several tissues besides the wing imaginal disc, such as the gut, the brain and the eye imaginal disc, by means of transgenesis, genome editing, transplantation and drug-induced toxic damage. These tumour models have provided new concepts and findings in cancer biology that have significant parallels and relevance to human cancer. Our findings establish the Drosophila wing imaginal disc as a platform to address the complex role of integrins in cancer. We can now extend these studies to the other Drosophila cancer models to further explore the multiple biological functions mediated by integrins in cancers.” And this is what we are currently doing, testing the role of integrins on modulating other cancer models, which include not only the the cooperation between RasV12 with loss of cell polarity, but also the overactivation of Notch or Yki pathways, using, besides the wing disc, other tissues, such as the eye imaginal disc, the brain and the gut. This requires a lot of work, including the generation of many fly stocks carrying all the necessary constructs to downregulate integrins and manipulate the different tumor-generating pathways in each specific tissue. Therefore, we believe is out of the scope of this paper and constitutes part of our future directions.

  1. In Fig 5F, the green should mean % of wing discs with GFP+ cells in the V domain.

Yes, sorry about this. This has now been corrected.

  1. The legends are also too short and unfriendly for non-fly readers. The authors should modify all the corresponding legends.

We have tried to modify the legends, explaining in more detail schemes and pictures, making them more friendly for non-fly readers.

Round 2

Reviewer 1 Report

Comments and Suggestions for Authors

Dear Dr Martin-Bermudo,

I enjoyed reading your manuscript. I find convincing your counter-argument and appreciate the lack of specific reagents and time to perform the experiment I suggested to do. I also find very useful the new citation (#32) that you added in the revised manuscript. I am happy to see this work published.

Best regards